# Prevalence of Select Intestinal Parasites in Alabama Backyard Poultry Flocks

**DOI:** 10.3390/ani11040939

**Published:** 2021-03-26

**Authors:** Miranda Carrisosa, Shanhao Jin, Brigid A. McCrea, Kenneth S. Macklin, Teresa Dormitorio, Rüdiger Hauck

**Affiliations:** 1Department of Poultry Science, College of Agriculture Auburn University, Auburn, AL 36849, USA; mcc0033@auburn.edu (M.C.); macklks@auburn.edu (K.S.M.); dormitv@auburn.edu (T.D.); 2College of Veterinary Medicine, Yangzhou University, Yangzhou 225009, Jiangsu, China; jinshanhao00@163.com; 3Alabama Cooperative Extension System, Auburn, AL 36849, USA; mccreba@auburn.edu; 4Department of Pathobiology, College of Veterinary Medicine, Auburn University, Auburn, AL 36849, USA

**Keywords:** epidemiology, Eimeria, coccidia, nematodes, zoonosis

## Abstract

**Simple Summary:**

As biosecurity is generally low in backyard chicken flocks, infections with various pathogens are common. This puts other poultry nearby, including commercial flocks, at risk. Some chicken pathogens can also infect humans and cause disease. In this study, backyard poultry flocks were tested for parasites. Eighty-four fecal samples, 82 from chickens and two from turkeys, from 64 backyard flocks throughout the state of Alabama were collected in the summers of 2017 and 2018. The most frequently observed parasites were coccidia, unicellular parasites capable of causing diarrhea. Eggs of various roundworms were observed in 20.3–26.6% of the flocks. These parasites were usually present in low numbers only. Other detected parasites were the flagellates *Histomonas meleagridis* and *Tetratrichomonas gallinarum* in 4.7% and 18.8% of flocks. Both can cause severe disease in poultry. Detected parasites that can cause disease in humans were *Cryptosporidium* spp. in 18.8% of the flocks and *Blastocystis* spp. in 87.5% of the flocks. The results will help to provide information that can be used to design outreach programs to improve the health and wellbeing of birds in backyard flocks.

**Abstract:**

Keeping chickens as backyard pets has become increasingly popular in the United States in recent years. However, biosecurity is generally low in backyard flocks. As a consequence, they can serve as reservoirs for various pathogens that pose a risk for commercial poultry or human health. Eighty-four fecal samples, 82 from chickens and two from turkeys, from 64 backyard flocks throughout the state of Alabama were collected in the summers of 2017 and 2018. Coccidia oocysts were seen in 64.1% of flocks with oocyst counts in most samples below 10,000 oocysts per gram. Eggs of *Ascaridia* spp. or *Heterakis gallinarum* were observed in 20.3% of the flocks, and eggs of *Capillaria* spp. in 26.6% of the flocks. Egg counts were low, rarely exceeding 1000 eggs per gram. DNA extracted directly from fecal samples was investigated by PCR for other relevant parasites. The results showed that 4.7% of flocks were positive for *Histomonas meleagridis*, 18.8% of flocks for *Tetratrichomonas gallinarum*, 18.8% of flocks for *Cryptosporidium* spp. and 87.5% of flocks for *Blastocystis* spp. The results will help to provide information that can be used to design outreach programs to improve health and wellbeing of birds in backyard flocks.

## 1. Introduction

Over the past two decades, there has been an apparent increase in backyard flocks in the United States [1,2,3,4,5,6]. “Backyard flock” is a term that generally refers to a privately owned flock of poultry, more often chickens than turkeys, that are kept at a residence. The most common reasons for backyard flock ownership in the United States are to keep the chickens as pets, a learning tool for children or as a source of eggs [3]. In other parts of the world, small non-commercial chicken flocks are referred to as village chickens and contribute to the subsistence of their owners [7].

Regardless of the location, many of these small flock owners tend to lack knowledge of proper biosecurity measures, e.g., wearing designated clothes/shoes, not allowing guests to interact with the chickens. They are not aware of the risks associated with exposing their flock to wild birds and rodents [3,8,9]. Zoonotic avian diseases such as salmonellosis are a risk for small flock owners, either by direct contact with backyard poultry flocks or by consumption of contaminated meat or eggs [10,11]. Low biosecurity in backyard flocks may also be an issue for commercial poultry flocks as backyard flocks can become a reservoir for pathogens [12]. This is especially relevant in a state like Alabama, which ranks second in broiler production in the United States [13].

*Eimeria* spp. are considered ubiquitous in chicken flocks [14]. However, their prevalence in village chickens can be between less than 5% [15,16] and up to more than 60% [17]. Roundworms, mostly *Ascaridia galli* and *Heterakis gallinarum*, have been detected in 15–25% of chickens by coproscopy [15,18] and up to 80% by visual inspection of intestines [19]. In 25% of dead village chickens, helminths were regarded as causative factor for the loss [20]. Currently, limited information is available about parasites found in backyard flocks in the United States. In birds submitted from backyard flocks to eight veterinary diagnostic laboratories across the United States, internal parasites were regarded as the primary cause of mortality in 2.6% of the birds. However, parasitic infections were the most common secondary finding, being observed in 25.5% of the birds [1].

The aim of the present study was to determine the population of relevant parasitic organisms found in backyard poultry flocks without ongoing disease.

## 2. Materials and Methods

### 2.1. Sample Collection

Eighty-four fecal samples from 64 different, non-commercial backyard flocks with less than 50 chickens throughout the state of Alabama were included in the study. The flocks were selected opportunistically, and pooled fecal samples of 10 to 50 g were collected in Ziploc bags and submitted by the owners. Forty-seven samples from 41 flocks were submitted in the summer of 2017 and 37 samples from 23 flocks in the summer of 2018. Two of the fecal samples were from turkeys kept on the same premises with sampled chickens. Each sample was stored at 4 °C upon arrival for microscopy and at −20 °C for DNA extraction. Four owners submitted samples of their flocks in both 2017 and 2018; however, in the present study they are considered different flocks.

### 2.2. Oocysts and Nematode Egg Detection

Each fecal sample was mixed thoroughly, and 1 g was suspended in 29 mL saturated NaCl solution. Debris was filtered out through a sieve. A McMaster chamber was filled with the fecal mixture and placed on a microscope where *Eimeria* spp. oocysts and nematode eggs were counted. The total number of oocysts and eggs in the chambers were multiplied by 100 to obtain the oocysts per gram (opg) and eggs per gram (epg) [21,22]. Eimeria oocysts were between 10 and 30 µm long and between 10 and 20 µm wide with a thick, double layered, smooth oocyst wall. Some oocysts were sporulated while most were not. *Ascaridia* spp. and *H. gallinarum* eggs were between 75 and 80 µm long and between 45 and 50 µm wide with a thick, smooth shell. *Ascaridia* spp. and *H. gallinarum* eggs were not differentiated due to similar egg morphology [23,24]. Capillaria eggs were about 70 µm long and 30 µm wide, with a thick smooth shell and two polar plugs.

### 2.3. Oocyst Purification and qPCR to Detect Eimeria

Oocysts were purified and concentrated from 4 g feces of 47 samples with Eimeria oocysts as described by Hafeez et al. [25]. Three positive samples were not further processed due to lack of material. DNA was extracted from the purified oocysts using the QIAGEN QiaAmp DNA mini kit (QIAGEN, Valencia, CA, USA) according to the manufacturer’s protocol, and Eimeria DNA was quantified by qPCR with 45 cycles detecting Eimeria 5S rDNA as described [26,27]. DNA load was expressed as the number of the cycles of the qPCR minus the quantification cycle (Cq). Its correlation with the parasite load in opg was assessed by calculating Spearman’s rho using R 3.6.0 [28].

### 2.4. Stool DNA Extraction and PCR for Other Parasites

DNA was extracted from one fecal sample per flock using the QIAGEN QIAamp Stool Mini Kit according to the manufacturer’s instructions (QIAGEN, Valencia, CA, USA). *Histomonas meleagridis*, *Tetratrichomonas gallinarum*, *Blastocystis* spp., and *Cryptosporidium* spp. and were detected by PCR using established protocols. Primers and references are listed in Table 1. Positive and negative controls were included in all PCR runs, and a negative control was included in all DNA extractions.

## 3. Results

### 3.1. Eimeria and Nematode Prevalence

Eimeria were detected in 41 flocks (64.1%) and 50 samples (59.5%). Median parasite load was 800 opg; however, several samples had greater than 10,000 opg. In the two samples from turkeys, no coccidia were observed. *Ascaridia* spp. or *H. gallinarum* eggs were detected in 13 flocks (20.3%) and 16 samples (19.0%), while *Capillaria* spp. were present in 17 flocks (26.6%) and 22 samples (26.2%). Median epg for all nematodes was less than 500 (Table 2, Figure 1). In one of the two samples from turkeys, 200 epg *Ascaridia* spp. or *H. gallinarum* eggs were observed. Of the four flocks that submitted samples in both 2017 and 2018, one flock had a change in status for coccidia from negative to positive and two had changes in Ascaridia/Heterakis egg status from negative to positive. There were no changes in status for *Capillaria* spp. eggs from year 2017 and 2018.

### 3.2. Quantification of Eimeria Oocysts by qPCR

The 5S rDNA qPCR failed to detect Eimeria DNA in four samples with 100 opg (two samples), 4000 opg, and 30,800 opg. A spearman’s rho of 0.31 showed only a weak correlation between the parasite load seen in the feces and the DNA load detected by qPCR (Figure 2).

### 3.3. Prevalence of Other Parasites

Of the 64 DNA samples, one from each flock, tested by PCR, 4.7% were positive for *H. meleagridis*, 18.8% for *T. gallinarum*, 18.8% for *Cryptosporidium* spp., and 87.5% for *Blastocystis* spp. (Table 2). Of the four flocks whose owners submitted samples in both 2017 and 2018, only one flock had a change in status for both *T. gallinarum* and *Blastocystis* spp. with the flock being positive in 2017 but negative in 2018. There were no changes in any of the other species of parasites in the flocks that submitted samples in both years.

## 4. Discussion

Backyard flocks may be a concern to public health and the commercial industry as they could potentially be a reservoir for pathogens. This is due to the fact that many of these flocks have poor biosecurity and have frequent access to the outdoors, which allows them to come in contact with wild birds and other animals, such as rodents, that can transmit disease [6,32].

In the present study, *Eimeria* spp. oocysts were detected in 59.5% of the samples and 64.1% of the flocks and counts in most samples were low. This reflected the equilibrium between infection and immunity present in older chickens, as well as the lower stocking density in extensively kept, often free ranging backyard flocks [3], which decreases the infection pressure. Compared to the prevalence of *Eimeria* spp. reported in village chickens, which ranges from less than 5% [15,16] up to more than 60% [17], this was comparatively high. One of the most important factors influencing the prevalence of *Eimeria* spp. in small flocks is the season [16,33,34]. In India, prevalence varied between 61% during monsoon season, i.e., warm and humid conditions, and 22% during the preceding cooler months [35]. The prevalence in the samples of this study taken during summer in Alabama was similar to the former number.

Four samples in which coccidia oocysts were observed, tested negative by qPCR. One likely reason for the discrepancy is a lack of sensitivity of the qPCR: two of the samples in question contained only 100 opg. On the other hand, two samples with considerably higher oocyst counts tested negative as well. The most likely reason is that the observed oocysts were not Eimeria infecting chickens but other coccidia, including *Eimeria* spp. infecting other hosts; the primers of this qPCR were designed based on sequences of Eimeria infecting chickens [26] and might not amplify other *Eimeria* spp. or coccidia. In fact, Eimeria from other hosts such as squirrels and mice were detected in some of the samples when amplified by pan-Eimeria PCR primers [27], potentially the result of coprophagy by chickens or contamination of the samples with feces from other hosts.

There was only a weak correlation between the parasite load seen in the feces and the DNA load detected by qPCR. The reasons probably include the presence of Eimeria from other hosts in addition to varying losses of oocysts during the purification of the oocysts and age of the samples. Testing samples from commercial poultry by the same methods showed a better correlation and no sample in which Eimeria oocysts had been seen tested negative by qPCR (results not shown).

A prevalence of 20% for eggs of *A. galli* or *H. gallinarum* and 26% for eggs of *Capillaria* spp. in the present study were similar to the prevalence of these parasites in village chickens in Africa when fecal samples were investigated [15,18]. However, the prevalence was lower than in organic layer chickens in Europe, where flock prevalence of the two parasites was between 49.3% and 100% [24]. However, the mean of 576 epg was similar to the results presented here [24].

Since coccidia and roundworms were the most encountered parasites in similar studies, they were our primary target and consequently flotation was used for detection. However, flotation might not be the most suitable method for tapeworm eggs, and we might have missed infections with those.

In European commercial pullet and layer flocks, antibodies against *H. meleagridis* were detected in up to 37.3% of the tested birds and 89.3% of the tested flocks [36,37]. In contrast, in the present study, the prevalence as detected by PCR was extremely low. This compares to findings by Cadmus et al. [1] who diagnosed histomoniasis based on lesions only in very few chickens. No nematode eggs were detected in the samples that tested positive for *H. meleagridis*. Unfortunately, these samples were very dry, which decreased the likelihood to detect nematode eggs.

In commercial poultry in Germany, *T. gallinarum* DNA was detected in 17.7% of flocks in which lesions resembling histomoniasis were observed, which is similar to the flock prevalence found here [38]. To our knowledge, there are no previous studies investigating its prevalence in commercial or backyard poultry without reported disease. In the present study, a single flock had a concurrent infection with both *T. gallinarum* and *H. meleagridis*.

Zoonotic parasites that were investigated included *Cryptosproidium* spp. and *Blastocystis* spp. There are several species of *Cryptosporidium* that are known zoonotic agents. *Cryptosporidium* spp. can cause intestinal disease in humans [39]. *Cryptosporidium meleagridis*, an avian pathogen, has been shown to be increasingly important as a human pathogen as it makes up 10–20% of human cryptosporidiosis cases in Peru and Thailand [40]. Due to the low host specificity of *C. meleagridis* and other *Cryptosporidium* spp., it is important for backyard flock owners to be aware of this and improve biosecurity as they could potentially become ill.

*Blastocystis* spp. are very common in chickens and seem to have a low host specificity. Blastocystis infections in humans may result in clinical symptoms such as diarrhea, abdominal cramps, and nausea. However, it is unclear if *Blastocystis* spp. infecting chickens can cause disease in humans [41,42]. In this study, there was a high prevalence of *Blastocystis* spp. with 87.5% of backyard flocks being infected. This is unsurprising as another study found a Blastocystis prevalence of 95% in commercial chickens [43]. 

Overall, only one flock was free of the parasites investigated and 11 flocks were only infected with a single parasite. Fifty-two flocks had concurrent infection with two or more of the selected parasites. However, prevalence of the investigated parasites in backyard flocks was lower than expected as chickens with access to the outdoors generally have higher rates of parasites [44,45].

The comparatively low prevalence of parasites is likely underestimated. Shedding patterns of the parasites may have an effect on the results. Eimeria are shed in variable amounts based on days post-infection [46]. Compared to our and similar results using coproscopy, detecting worms macroscopically during necropsy resulted in significantly higher prevalence of more than 65% [19,47]. A study that looked into diurnal fluctuations of nematode egg excretion found that egg shedding was higher during the day, early morning to noon, than in the afternoon and night [48], and *H. meleagridis* is shed only intermittently from chronically infected chickens [49]. In addition, sample quality was not always optimal due to flock owners collecting samples and not properly storing them; and for some parasites the McMaster flotation method using 1g of feces, especially of pooled samples, may be too insensitive to show all present dispersive forms of parasites.

## 5. Conclusions

We detected a variety of poultry parasites in the investigated flocks, posing a risk to commercial poultry and their owners. The results of this study will help to provide information to owners of backyard chicken flocks that can be used to design timely and appropriate extension/outreach material. Informing them of the types of internal parasites typically observed and what steps can be implemented to improve the health and wellbeing of birds will improve the overall health of backyard flocks. In addition, alerting flock owners of potential zoonotic parasites that are present in their backyard flock may lead to improvements in their biosecurity measures.

## Figures and Tables

**Figure 1 animals-11-00939-f001:**
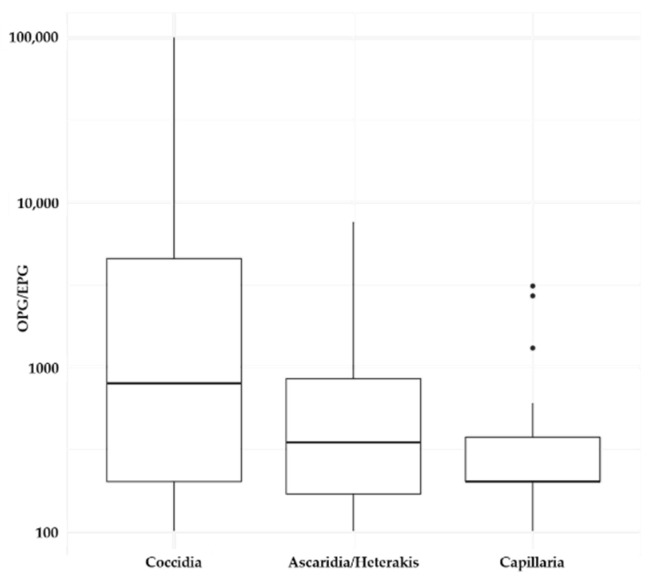
Coccidia oocysts per gram feces (OPG) and *Ascaricdia galli*/*Heterakis gallinarum* eggs per gram feces (EPG) detected in fecal samples of backyard chicken flocks shown on log scale.

**Figure 2 animals-11-00939-f002:**
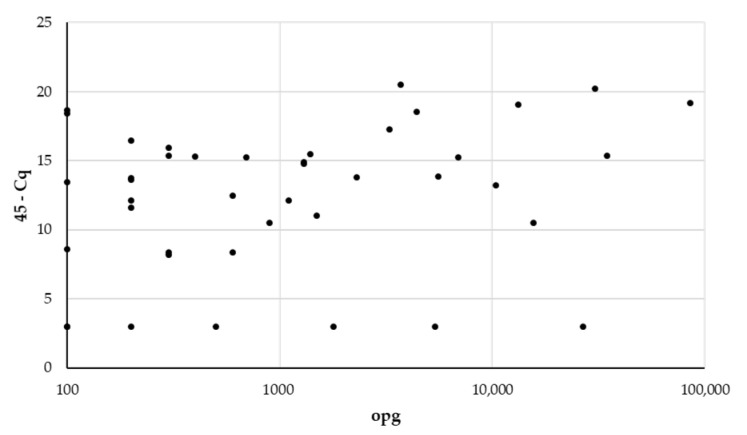
Scatterplot correlating the number of coccidia oocysts per gram feces (opg) and the detected DNA load expressed as the number of the cycles of the qPCR minus the quantification cycle Cq.

**Table 1 animals-11-00939-t001:** Nucleotide sequences and references of the primers used to detect the parasites in fecal samples of backyard chicken flocks.

Species	Sequences		Amplicon Size in Base Pairs	Reference
	Forward (5’-3’)	Reverse (5’-3’)		
*Eimeria* spp.	TCA TCA CCC AAA GGG ATT	TTC ATA CTG CGT CTA ATG CAC	~110	[26]
	Probe: [6-FAM] CGC CGC TTA ACT TCG GAG TTC AGA TGG GAT [BHQ-1] ^1^			
*Blastocystis* spp.	TAA CCG TAG TAA TTC TAG GGC	AAC GTT AAT ATA CGC TAT TGG	459	[29]
*Cryptosporidium* spp. (outer)	TTC TAG AGC TAA TAC ATG CG	CCC TAA TCC TTC GAA ACA GGA	1325	[30]
*Cryptosporidium* spp. (nested)	GGA AGG GTT GTA TTT ATT AGA TAA AG	AAG GAG TAA GGA ACA ACC TCC A	830	[30]
*Histomonas meleagridis*	CCG TGA TGT CCT TTA GAT GC	GAT CTT TTC AAA TTA GCT TTA AAT TAT TC	603	[31]
*Tetratrichomonas gallinarum*	GCA ATT GTT TCT CCA GAA GTG	GAT GGC TCT CTT TGA GCT TG	526	[29]

^1^ 6-Carboxyfluorescein; Black Hole Quencher.

**Table 2 animals-11-00939-t002:** Prevalence of *Eimeria* spp. oocysts and nematodes eggs in fecal samples of backyard chicken flocks and median oocysts (OPG) and eggs per gram (EPG).

Species	Positive Samples (*n* = 84)	Positive Flocks (*n* = 64)	Median opg/epg
*Eimeria* spp.	50 (59.5%)	41 (64.1%)	800
*Ascaridia galli*/*Heterakis gallinarum*	16 (19%)	13 (20.3%)	350
*Capillaria* spp.	22 (26.2%)	17 (26.6%)	200
*Histomonas meleagridis*	n.a. ^1^	3 (4.7%)	n.a. ^2^
*Tetratrichimonas gallinarum*	n.a.	12 (18.8%)	n.a.
*Cryptosporidium* spp.	n.a.	12 (18.8%)	n.a.
*Blastocystis* spp.	n.a.	56 (87.5%)	n.a.

^1^ not applicable because only one sample per flock was investigated ^2^ not applicable because no quantitative test.

## Data Availability

All data is contained within the article.

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
