# Peer review of "Prevalence of Select Intestinal Parasites in Alabama Backyard Poultry Flocks"

_animals, 2021, doi:10.3390/ani11040939_

Round 1

Reviewer 1 Report

The manuscript is well-written and contains important scientific information on the prevalence of important internal parasites in backyard flocks. However, please address the following comments.

Specific comments

Lines 14-15: Because biosecurity is generally low in backyard chicken flocks and infections with various pathogens are common.

Replace ‘and’ with a coma.

Lines 15-16: This puts other poultry nearby, including commercial flocks, at risks.

Change to ‘at risk’.

Lines 55-56: Low biosecurity in back-yards flocks may also

Change to ‘back-yard’

Lines 82-89: Oocysts and nematode egg detection.

Briefly describe the morphological features/criteria used to determine Eimeria oocysts and nematode eggs.

Figure. 1: Please increase the font size. The labeling on x and y axis are not clear.

Lines 156-160: One of the most important factors influencing the prevalence of Eimeria spp. in small flocks is the season [16,33,34]. In India, prevalence varied between 61% during  monsoon season, i.e. warm and humid conditions, and 22% during the preceding cooler months [35]. The prevalence in this study is in agreement with the former number, which is reasonable considering that all samples were collected during summer in Alabama, which means in warm and humid weather.

This reasoning is weak. The weather in India during Monsoon is entirely different from summer weather in Alabama. Please change/address.

Author Response

Thank you for the positive and constructive comments that have helped us to improve the manuscript. Please find our responses below. In brief, we followed all your recommendations.

The manuscript is well-written and contains important scientific information on the prevalence of important internal parasites. However, please address the following comments.

Specific comments

Lines 14-15: Because biosecurity is generally low in backyard chicken flocks and infections with various pathogens are common.

Replace ‘and’ with a coma.

We changed to “Because biosecurity is generally low in backyard chicken flocks, infections with various pathogens are common.” (line 14)

Lines 15-16: This puts other poultry nearby, including commercial flocks, at risks.

Change to ‘at risk’.

We deleted the “s.” (line 16)

Lines 55-56: Low biosecurity in back-yards flocks may also

Change to ‘back-yard’

We deleted the “s.” (line 56)

Lines 82-89: Oocysts and nematode egg detection.

Briefly describe the morphological features/criteria used to determine Eimeria oocysts and nematode eggs.

We added the information. The whole paragraph now reads “Each fecal sample was mixed thoroughly, and 1 g was suspended in 29 mL saturated NaCl solution. Debris was filtered out through a sieve. A McMaster chamber was filled with the fecal mixture and placed on a microscope where Eimeria spp. oocysts and nematode eggs were counted. The total number of oocysts and eggs in the chambers were multiplied by 100 to obtain the oocysts per gram (opg) and eggs per gram (epg). [21,22]. Eimeria oocysts were between 10 and 30 µm long and between 10 and 20 µm wide with a thick, double layered, smooth oocyst wall. Some oocysts were sporulated while most were not. Ascaridia spp. and H. gallinarum eggs were between 75 and 80 µm long and between 45 and 50 µm wide with a thick, smooth shell. Ascaridia spp. and H. gallinarum eggs were not differentiated due to similar egg morphology [23,24]. Capillaria eggs were about 70 µm long and 30 µm wide, with a thick smooth shell and two polar plugs.” (lines 83 – 95)

Figure. 1: Please increase the font size. The labeling on x and y axis are not clear.

We relabeled the axes with a larger font size.

Lines 156-160: One of the most important factors influencing the prevalence of Eimeria spp. in small flocks is the season [16,33,34]. In India, prevalence varied between 61% during monsoon season, i.e. warm and humid conditions, and 22% during the preceding cooler months [35]. The prevalence in this study is in agreement with the former number, which is reasonable considering that all samples were collected during summer in Alabama, which means in warm and humid weather.

This reasoning is weak. The weather in India during Monsoon is entirely different from summer weather in Alabama. Please change/address.

Monsoon in India and summer in Alabama are both warm and humid, e.g. Hyderabad average temperature between 25 and 28C, 101 – 162 mm rainfall from June to August and Montgomery average temperature between 27 and 29C, 99 – 105 mm rainfall in the same months (https://en.climate-data.org/north-america/united-states-of-america/alabama/montgomery-1602/ and https://en.climate-data.org/asia/india/hyderabad/hyderabad-2801/). However, we understand that comparisons like this are always imperfect. We rephrased the sentences to avoid a direct comparison of the weather “One of the most important factors influencing the prevalence of Eimeria spp. in small flocks is the season [16,33,34]. In India, prevalence varied between 61% during monsoon season, i.e. warm and humid conditions, and 22% during the preceding cooler months [35]. The prevalence in the samples of this study taken during summer in Alabama, was similar to the former number.” (lines 166 – 169)

Reviewer 2 Report

Ewaluation of sceientifc work in the atached file. 

Author Response

We thank the reviewer for the positive but especially the critical comments. Even while we do not fully agree with all of them, as outlined below, they have helped us to improve this manuscript and will help us in the future with further research.

The assumption of the research work and its goals are very interesting.

The selection of research methods, especially when it comes to microscopic examination of stools, in my opinion is inappropriate. For this reason, the authors obtained a surprisingly low parasite diversity, which does not indicate the actual state of infestation in chickens. It is surprising that the infestation of tapeworms and many other nematodes, such as the genera Syngamus, Trichostrongylus, Amidostomum, and Spiruriden, has not been shown.

We are aware that by using sedimentation in addition to flotation we might have found tapeworm eggs. However, just like for the parasites detected by PCR, we had to limit ourselves for practical reasons and could only test for selected parasites. We acknowledge this now in the discussion “Since coccidia and roundworms were the most encountered parasites in similar studies, they were our primary target and consequently flotation was used to detection. However, flotation might not be the most suitable method for tapeworm eggs, and we might have missed infections with those.” (lines 192 – 195)

The eggs of other roundworms would have been detected by flotation. Our results that Ascaridia/Heterakis and Capillaria are the most frequently encountered species is in agreement with most other similar studies we cite, so not detecting other nematodes was not entirely surprising for us.

Also the obtained results for Ascaridia, Heterakis and Eimeria are therefore not very reliable. In the Mc Master's method, 1 g of faeces was tested. Considering that the samples came from many birds, the (collective) representativeness of the 1g sample is small. Therefore, I suspect that many of the invasions have not been detected. These results would be reliable if single (individual) samples of bird feces were tested and, in addition to the quantitative Mcmaster method (2 g of sample), the classical flotation method was used with at least 3 g or larger fecal samples.

The 1 g aliquots for the Mcmaster method were taken from well mixed composite samples that were in most cases much larger (line 75). Arguably, taking a 2 g aliquot might have slightly increased the representativeness, even though the actual amount in the chamber, i.e. the fraction of the total sample looked at under the microscope would have been the same. McMaster chambers have been used as the sole microscopic method in many current studies investigating field samples from poultry flocks for Eimeria spp. and worms.

With regard to the genetic detection of Eimeriosis, unrepresentative primers were probably

used.

The reference for the primer-probe combination is Blake, D.P.; Hesketh, P.; Archer, A.; Shirley, M.W.; Smith, A.L. Eimeria Maxima: The influence of host genotype on parasite reproduction as revealed by quantitative real-time PCR. Int J Parasitol 2006, 36, 97–105, doi:10.1016/j.ijpara.2005.09.011. The PCR has been validated to detect all known Eimeria spp. infecting chickens. By definition, it is not possible to know if they can detect unknown species. It is likely that this primer-probe combination also detects some Eimeria spp. infecting other hosts. It would be very difficult or even impossible and of minor importance for the present study to determine which other Eimeria spp. are detected by this primer-probe combination.

The explanation that alien species have not been detected is unreliable to me. With such a high OPG (Eimeria spp.), it is hard to believe that they were alien, random species. With such a high OPG, they must have multiplied. However, they could be species typical for chickens, but with low pathogenicity, not included in molecular clinical diagnostics.

The speciation by ngs amplicon sequencing in the cited paper (Hauck, R.; Carrisosa, M.; McCrea, B.A.; Dormitorio, T.; Macklin, K.S. Evaluation of next-generation amplicon sequencing to identify Eimeria spp. of chickens. Avian Dis 2019, 63, 577–583, doi:10.1637/aviandiseases-D-19-00104) using these very samples shows that there were Eimeria spp. from other hosts and not unknown Eimeria species infecting chickens. However, it might be possible that the contamination is not (only) by coprophagy, but that the owners contaminated the samples with feces from other hosts. We now mention this possibility: “…potentially the result of coprophagy by chickens or contamination of the samples with feces from other hosts.” (line 179)

The work also has very valuable elements - the results of genetic research, especially in relation to zoonotic invasions. These results can be the basis of a very interesting work showing the threat to human health. This work, however, requires a change of title and a thorough redrafting.

Thank you for the kind remark regarding the value of our study. However, if we understand correctly, this is a suggestion to delete all results relating to Eimeria and nematodes. While we acknowledge that the inclusion of sedimentation might have allowed us to detect more parasites and that investigating larger aliquots might have yielded slightly more representative counts, we are convinced that the results are valid and do not see how deleting them would improve the manuscript.

Reviewer 3 Report

The manuscript is much improved

Author Response

Thank you for this positive comment and, presumably, for helpful comments on a previous version of this manuscript.

Round 2

Reviewer 2 Report

Opinion of a revised scientific work

The proposed changes can be accepted.

In the discussion, perhaps in the conclusions of the work, in my opinion there is still no explanation that for some invasions the McMaster flotation method using 1g of faeces (especially in the case of aggregate samples) may be too insensitive (a larger mass of the tested sample should be used in the future) to show all present dispersive forms of parasites.

Author Response

This is a very fair comment. We added “for some parasites the McMaster flotation method using 1g of feces, especially of pooled samples, may be too insensitive to show all present dispersive forms of parasites”, which is almost verbatim the comment, in lines 231 – 233 of the manuscript, and will keep this in mind for further studies.